# Global huntingtin knockout in adult mice leads to fatal neurodegeneration that spares the pancreas

Robert M Bragg[1,2,]*, Ella W Mathews[1,2,]*, Andrea Grindeland[3], Jeffrey P Cantle[1], David Howland[4], Tom Vogt[4], Jeffrey B Carroll[1,2]

Huntington's disease (HD) is a fatal neurodegenerative disorder caused by an expanded CAG tract in the huntingtin (HTT) gene, leading to toxic gains of function. HTT-lowering treatments are in clinical trials, but the risks imposed are unclear. Recent studies have reported on the consequences of widespread HTT loss in mice, where one group described early HTT loss leading to fatal pancreatitis, but later loss as benign. Another group reported no pancreatitis but found widespread neurological phenotypes including subcortical calcification. To better understand the liabilities of widespread HTT loss, we knocked out *Htt* with two separate tamoxifen-inducible Cre lines. We find that loss of HTT at 2 mo of age leads to progressive tremors and severe subcortical calcification at examination at 14 mo of age but does not result in acute pancreatitis or histological changes in the pancreas. We, in addition, report that HTT loss is followed by sustained induction of circulating neurofilament light chain. These results confirm that global loss of HTT in mice is associated with pronounced risks, including progressive subcortical calcification and neurodegeneration.

## Introduction

The huntingtin (*HTT*) gene is widely conserved and ubiquitously expressed across investigated cell types. *HTT* is most well known as the gene in which an expansion of a glutamine-coding CAG repeat causes Huntington's disease (HD), an autosomal dominant, fatal, neurodegenerative disease (1). *HTT* expression appears to be very important, particularly during development, as *Htt* null mice die very early in embryogenesis (2, 3, 4), and compound heterozygous expression of hypomorphic *HTT* alleles causes profound neuro-developmental impairment in humans, with non-progressive features distinct from HD (5, 6). Population databases reveal that putative loss of function mutations in *HTT* are highly selected

against (e.g., the abundance of observed/expected putative loss of function mutations in the genome aggregation database (gnomAD (7)) is = 0.12; pLI score = 1).

Genetic evidence suggests that HD pathology primarily arises from a toxic gain of function (1). This, coupled with dominant and complete penetrance, makes huntingtin (HTT)-lowering therapies an attractive approach to HD treatment (8). Indeed, a wide range of HTT-lowering approaches have already advanced to human clinical studies, including ongoing studies with antisense oligonucleotides (ASOs), virally delivered microRNA (miRNAs), and most recently small molecule splice modulators that favor the inclusion of a cryptic exon with a premature stop codon, thereby reducing HTT levels across the body (9, 10). Some of these approaches are inherently allele-selective, only targeting mutant HTT (mHTT), for example, intrathecally delivered ASOs developed by Wave Life Sciences which target single nucleotide polymorphisms in the *HTT* gene (11). Other approaches, for example, tominersen—an ASO targeting both wildtype and mutant *HTT*, currently being developed by Ionis Pharmaceuticals and Roche (11, 12)—target both alleles. Genetic evidence suggests that 50% HTT expression is well -tolerated in both mice and humans—the heterozygous parents of patients impacted by a syndrome caused by expression of very low levels of HTT are not described to have any clinically meaningful impact on their near 50% HTT expression (13). Similarly, in mice, whereas complete KO of *Htt* results in embryonic lethality, heterozygous KO mice are grossly normal—to our knowledge, the only reported phenotype in these mice is a reduction in body weight (14). If HTT-lowering therapies work, a key question about their safety is the extent to which HTT silencing is tolerated and the precise level of HTT required for healthy cellular function.

A key piece of information needed to assess the safety of HTT lowering is the tolerability of wide-spread, adult-onset HTT lowering. Indeed, two groups have published studies with similar designs that investigated the impact of tamoxifen-Cre-induced recombination and global HTT lowering in *Htt*-flox mice (15, 16). Of these, Dietrich et al (2017), found widespread deleterious effects of adult-onset HTT lowering, including progressive neurodegeneration accompanied by

---

[1]Department of Psychology, Western Washington University, Bellingham, WA, USA [2]Department of Neurology, University of Washington, Seattle, WA, USA [3]McLaughlin Research Institute, Great Falls, MT, USA [4]CHDI Foundation, Princeton, NJ, USA

Correspondence: jeffcarr@uw.edu
*Robert M Bragg and Ella W Mathews contributed equally to this work

thalamic calcification (16). In contrast, Wang et al (2016) found that global HTT silencing at 4 and 8 mo was benign through 18 mo of age, but that silencing HTT at 2 mo of age led to acute and fatal pancreatitis, with 95% fatality by 10 d of *Htt* loss (15). Motivated to better understand these apparently discordant results, we conducted a replication study.

We designed our study to closely mimic the two previous ones, with several additional assessments. We generated two separate cohorts of *Htt*-flox mice crossed with unique ubiquitous tamoxifen-inducible Cre recombinase lines. In the floxed *Htt* allele, Cre expression deletes the entire first exon, including the transcription start site, promoter, and part of intron 1. We first evaluated the leakiness—the tendency to recombine the targeted locus in the absence of tamoxifen treatment—in both Cre lines across a wide range of tissues using quantitative *Htt* mRNA and protein assays. We then initiated tamoxifen treatment to remove HTT expression beginning at 2 mo of age and lasting until sacrifice at 14 mo of age. In contrast to Wang et al, we did not observe pancreatitis after early-onset HTT loss. In addition to behavioral and pathological endpoints following extended HTT silencing, we also collected longitudinal plasma samples to measure levels of neurofilament light chain (NEFL)—a widely used biomarker for neuronal stress (17). Strikingly, we observed progressive neurological changes including the appearance of calcified deposits in the thalamus, confirming findings reported by Dietrich et al, and discovered that plasma NEFL is robustly increased by HTT loss.

In summary, we have conducted a replication study to establish the impact of widespread adult-onset HTT lowering. Our results are largely consistent with those of Dietrich et al (2017), whereas we observe significant differences from those of Wang et al (2016), notably an absence of acute fatal pancreatitis after early *Htt* deletion. Our results support the hypothesis that complete loss of *Htt* in adult animals is poorly tolerated, as it is associated with a range of phenotypes including progressive neurodegeneration accompanied by thalamic calcification.

# Results

## Establishment of allele use

Both previous studies relied on the B6.Cg-Tg(CAG-cre/Esr1*)5Amc/J, referred to as "CAG-Cre" mice (strain #004682; Jackson labs), in which a tamoxifen-inducible Cre construct is under the control of a ubiquitous promoter, leading to widespread recombination of floxed alleles after tamoxifen treatment (18). We also used a second ubiquitously expressed tamoxifen-inducible Cre line, which has been shown to have widespread efficiency, including in the brain (19) (B6.Cg-Ndor1^Tg(UBC−cre/ERT2)1Ejb/1^J), referred to as "UBC-Cre" (20). Both lines were crossed to a previously described floxed *Htt* allele, B6.Htt^tm2Szi^ (4), referred to as *Htt^fl/fl^*, resulting in multiple large cohorts of *Htt^fl/fl^*;CAG-Cre, *Htt^fl/fl^*;UBC-Cre and control lines, *Htt^+/+^* and *Htt^fl/fl^*, on a consistent C57Bl/6J background (Table 1), referred to here as conditional *Htt* knock-out (cKO) mice. In concordance with Wang et al, we maintained full HTT expression through development by using homozygous *Htt* flox alleles, rather than the approach used by Dietrich et al where the *Htt* flox allele was

crossed to *Htt* null allele. One additional consideration is that we backcrossed the *Htt^fl/fl^* mouse strain to B6 and confirmed congenicity with a 384 single nucleotide polymorphism panel before generating the cohorts described below (21). It is unclear if Dietrich et al and Wang et al confirmed the genetic background of their cohorts.

We next investigated the leakiness of each Cre construct in the absence of tamoxifen from 5–12 mo of age by quantifying the levels of HTT and the estimated percentage recombination frequency using electro-chemiluminescent ELISA and semi-quantitative PCR, respectively (Table 1). In general, our protein and DNA assays are consistent throughout this study. Interestingly, we observed that the *Htt^fl/fl^*;UBC-Cre line was extremely leaky at the floxed *Htt* locus in the CNS organs examined, with up to 83% reduction in HTT levels in the cortex at 12 mo of age (Fig 1B, Tukey's HSD, *P* < 0.0001), but was much more tightly regulated in the examined peripheral organs. In contrast, the *Htt^fl/fl^*;CAG-Cre exhibited more leak in peripheral organs, reaching up to a 53% reduction in HTT in the liver (Fig 1B, Tukey's HSD, *P* = 0.002), but was very tightly regulated in CNS tissue. A concordant phenomenon is observed where the floxed *Htt* region is recombined in the absence of tamoxifen in central tissues of the *Htt^fl/fl^*;UBC-Cre but peripheral tissues of the *Htt^fl/fl^*;CAG-Cre line (Fig S1A). We replicated these observations in a separate follow-up cohort of mice in which we took a wider tissue survey at a single time point (cortex, striatum, cerebellum, spinal cord, liver, kidney, pancreas, and gastrocnemius at 5 mo of age; Fig S2).

## Establishment of cohorts, study design, survival analysis

Based on the observations that *Htt^fl/fl^*;CAG-Cre and *Htt^fl/fl^*;UBC-Cre mice are differentially leaky across tissue types, we generated large cohorts of mice with both constructs to follow longitudinally. We chose to deploy the *Htt^fl/fl^*;CAG-Cre and *Htt^fl/fl^*;UBC-Cre mice for experiments looking for phenotypes arising in CNS, and peripheral organs, respectively. The experimental cohort included 160 mice—10M/10F from each genotype/treatment arm (Table 2) and a smaller group of mice for planned interim assessment at 3, 6, and 9 mo of age to monitor HTT loss across tissues during the study (Table 3). Following best practices established for tamoxifen treatment by Jackson Laboratories (22), we treated mice in our tamoxifen cohorts with 75 mg/kg tamoxifen for five consecutive days at 2 mo of age. The study design outlined in Fig 1A indicates the planned interventions and their timing. Based on previous findings by Wang et al (2016) (15), we were concerned that our mice would die from acute pancreatitis in response to HTT loss at 2 mo of age; however, we saw no acute death in any of our mice. As expected, our interim silencing cohorts at 3, 6, and 9 mo show very robust HTT loss compared with vehicle controls in the cortex, striatum, and liver (Fig 1C, Tukey HSD *P* < 0.01 for all tissues) and clear Cre-induced recombination across the floxed *Htt* allele (Fig S1A and B). In the pancreas, HTT is significantly lowered at 6 mo compared with vehicle control (Tukey HSD *P* < 0.001), but not at 3 or 9 mo, likely because of high variability in this tissue. By 12 mo, both control and cKO mice began to reach humane endpoint criteria because of treatment-resistant ulcerative dermatitis, whereas only cKO mice also displayed severe progressive tremor. We chose to euthanize all remaining mice at 14 mo of age to collect a sufficiently sized cohort

**Table 1. Leak assessment cohort.**

| Genotype | Sex | Treatment | Collection age (mo) | | | | | | | | |
|----------|-----|-----------|---|---|---|---|---|---|---|---|---|
| | | | 5 | 6 | 7 | 8 | 9 | 10 | 11 | 12 | |
| $Htt^{fl/fl}$ | M | Untreated | 5 | 5 | 5 | 5 | 5 | 5 | 5 | 5 | |
| $Htt^{fl/fl}$ | F | Untreated | 5 | 5 | 5 | 5 | 5 | 5 | 5 | 5 | |
| $Htt^{fl/fl;}$CAG-cre | M | Untreated | 5 | 5 | 5 | 5 | 5 | 5 | 5 | 5 | N |
| $Htt^{fl/fl;}$CAG-cre | F | Untreated | 5 | 5 | 5 | 5 | 5 | 5 | 5 | 5 | |
| $Htt^{fl/fl;}$UBC-cre | M | Untreated | 5 | 5 | 5 | 5 | 5 | 5 | 5 | 5 | |
| $Htt^{fl/fl;}$UBC-cre | F | Untreated | 5 | 5 | 5 | 5 | 5 | 5 | 5 | 5 | |

of age-matched tissue for robust pathological and molecular characterization.

## Longitudinal plasma chemistry and NEFL

Dietrich et al (2017) reported that ubiquitous *Htt* loss leads to progressive neurodegeneration, including thalamic calcification (16). In hopes of determining when this process begins, we performed longitudinal plasma sampling from five animals per arm with matched WT controls to quantify the levels of NEFL, a well-established marker of axonal stress, whose use as a translational biomarker has been established in many neurodegenerative disease states (17). We collected plasma beginning at 3 mo of age, continuing monthly until the study was complete at 14 mo of age. In the $Htt^{fl/fl}$;CAG-Cre line, we find that tamoxifen treatment leads to a rapid and persistent elevation of plasma NEFL levels throughout the duration of the study (Fig 2A). At 3 mo of age (1-mo post-tamoxifen treatment), mean plasma NEFL levels in tamoxifen-treated $Htt^{fl/fl}$;CAG-Cre mice (2,545 ± 678 pg/ml) are significantly increased compared with vehicle-treated $Htt^{fl/fl}$;CAG-Cre mice (840 ± 311 pg/ml; Tukey HSD $P$ < 0.0001), both of which are higher than vehicle- or tamoxifen-treated $Htt^{+/+}$ mice, which are below 300 pg/ml (Fig 2B; Tukey HSD $P$ < 0.0001). In the $Htt^{fl/fl}$;UBC-Cre mice, we find a slightly different pattern. Initially, we $Htt^{fl/fl}$;UBC-Cre are very similar to the $Htt^{fl/fl}$;CAG-Cre mice–plasma NEFL levels are significantly elevated at 3-mo of age in tamoxifen-treated mice (2,091 ± 914 pg/ml), compared with vehicle (719 ± 165 pg/ml; Fig 2B; Tukey HSD $P$ < 0.0001). However, in the vehicle-treated $Htt^{fl/fl}$;UBC-Cre mice, NEFL levels increase steadily to 3,467 ± 647 pg/ml, rising to levels similar to tamoxifen-treated mice (4,430 ± 953 pg/ml) by 13-mo of age (Fig 2A). We hypothesize that this is driven by the much leakier nature of Cre in the CNS of the $Htt^{fl/fl}$;UBC-Cre line, compared with the $Htt^{fl/fl}$;CAG-Cre line (Fig 1B), driving continued loss of HTT in the CNS during this period. From our final endpoint mice, we also conducted a cross-sectional screen of 11 plasma analytes and found that alanine transaminase, a biomarker of liver damage, was slightly elevated in tamoxifen-treated $Htt^{fl/fl}$;UBC-Cre mice compared with vehicle (Fig S3; Tukey HSD $P$ = 0.001).

## Behavioral analyses

Mice underwent a monthly modified SHIRPA examination to monitor neurological signs and several cross-sectional behavioral assays at 11 mo of age (Table S1). We observed progressive tremor

during the SHIRPA examination (Fig 3A) in both cKO lines starting at 3 mo and increasing in severity through 13 mo. In conjunction, we observed limited and abnormal gait in some of the cKO mice, but this was not as consistent. All other SHRIPA measures appeared normal. These finding largely match those of Dietrich et al who observed tremors and gait abnormalities, although we did not detect hindlimb clasping or pronounced kyphosis. For additional motor tasks, we excluded $Htt^{fl/fl}$;UBC-Cre because of less controlled HTT knockdown in CNS tissue. As previously reported (16), we observed motor impairment, as revealed by an elevated balance beam traversal assay in which tamoxifen-treated $Htt^{fl/fl}$;CAG-Cre mice take longer than vehicle-treated littermates (Fig 3B; $Htt^{fl/fl}$;CAG-Cre tamoxifen vs. $Htt^{fl/fl}$;CAG-Cre vehicle; Tukey HSD $P$ < 0.0001). We also conducted an examination of the total movement using an open field assay, revealing a potential sustained impact of tamoxifen treatment, rather than HTT loss, as tamoxifen treatment of both $Htt^{fl/fl}$ and $Htt^{fl/fl}$;CAG-Cre mice leads to modest hypoactivity (Fig 3C; $Htt^{fl/fl}$ tamoxifen vs. $Htt^{fl/fl}$ vehicle Tukey HSD $P$ = 0.02; $Htt^{fl/fl}$;CAG-Cre tamoxifen vs. $Htt^{fl/fl}$;CAG-Cre vehicle Tukey HSD $P$ = 0.04). In summary, we found that HTT loss is associated with progressive neurological alterations, as revealed by increasingly severe tremors and motor impairment.

## Terminal HTT and NEFL levels

Our interim analyses of HTT levels (Fig 1) and NEFL (Fig 2) were based on relatively small numbers of animals. At 14-mo of age we collected a more robust dataset from the complete endpoint cohort of mice, providing an opportunity to collect both cerebrospinal fluid (CSF) and plasma to directly compare NEFL levels in both compartments. In general, our HTT assays reveled a pattern consistent with our interim HTT lowering data (compare Figs 1C and 4A)—namely that the $Htt^{fl/fl}$;CAG-Cre line was relatively well-regulated in the CNS, whereas the $Htt^{fl/fl}$;UBC-Cre line was leaky in the CNS but more tightly regulated in the periphery (Fig 1B). Similarly, both the CSF and plasma NEFL quantifications revealed patterns consistent with those observed in our longitudinal study (compare Figs 2A and 4B).

## Peripheral athology

Given that early excision of *Htt* before 3 mo of age had been reported to cause fatal pancreatitis, we were particularly interested in the impact of HTT loss at 2 mo of age on the pancreas. We examined

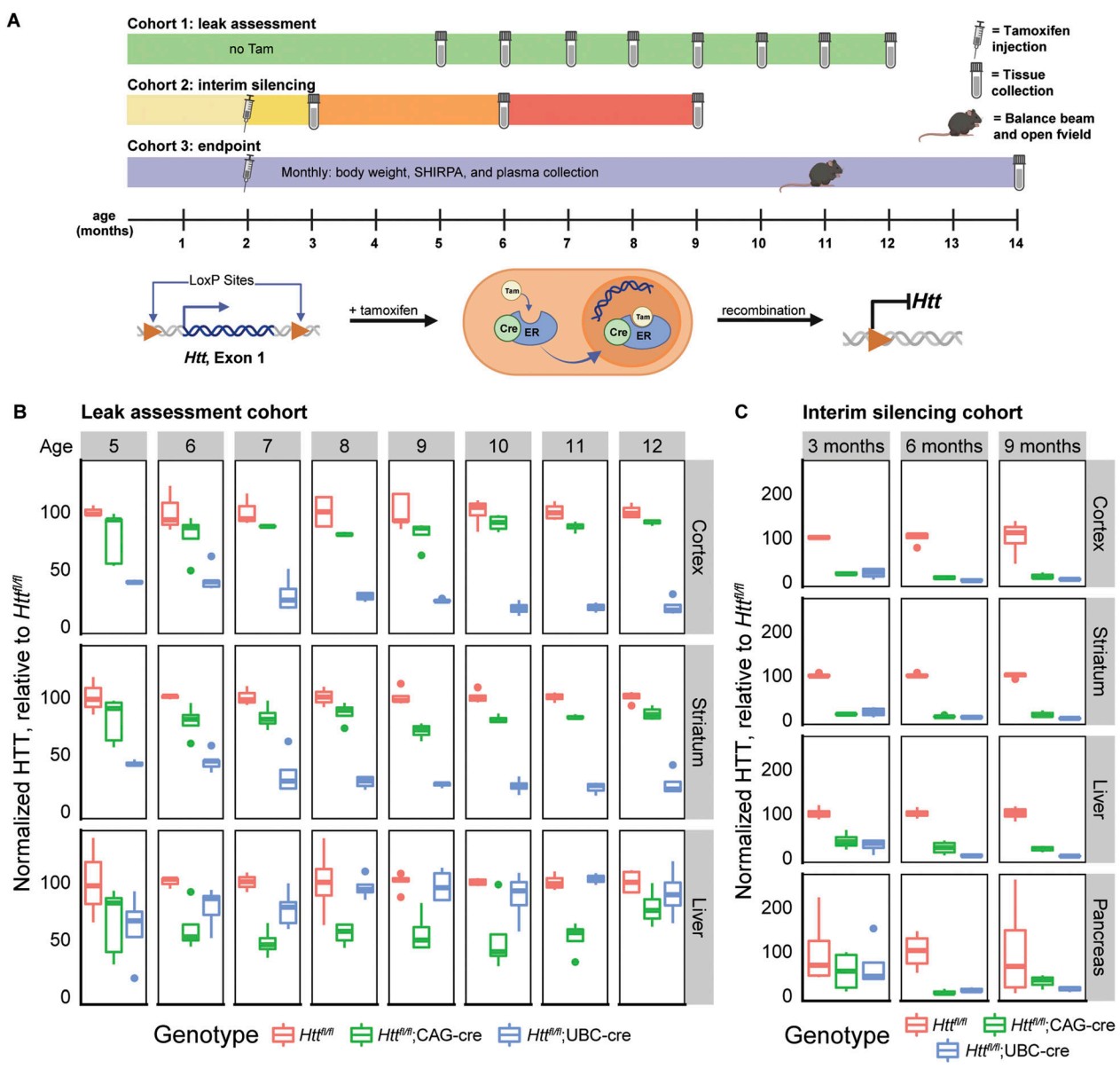

**Figure 1. Study design and mouse validation.**
**(A)** (Top) Timelines of experimental cohorts used in this study with indications of tamoxifen injection time, in-life analyses, and tissue collection points. (Bottom) Cartoon showing mechanism of *Htt* inactivation which occurs when tamoxifen (Tam) is administered and causes cytosolic cre-recombinase-estrogen receptor complex to translocate to the nucleus inducing recombination of the floxed Htt allele. **(B)** HTT levels measured in the absence of tamoxifen reveal differential leakiness of each Cre line across three tissues. n = 5/genotype/age, a full list of statistical comparisons available in Supplemental Data 1. **(C)** Levels of HTT are robustly and consistently reduced in tamoxifen-treated mice. Silencing in tamoxifen-treated animals at 3, 6, and 9 mo of age (columns) was assessed across a range of organs (rows). n = 4/group, a full list of statistical comparisons available on Supplemental Data 1. HTT levels in (B, C) were assayed using chemiluminescent ELISA and normalized to control group ($Htt^{fl/fl}$ mice).
Source data are available for this figure.

the amount of HTT loss in the pancreas across our lines—as with other peripheral organs, we found that the $Htt^{fl/fl}$;CAG-Cre mice were slightly leaky, whereas the $Htt^{fl/fl}$;UBC-Cre line much less so (Fig S1C). After tamoxifen treatment, we observe significant lowering at 6 mo ($Htt^{fl/fl}$;CAG-Cre compared with $Htt^{fl/fl}$: 93% reduction, Tukey HSD $P$ = 0.002; $Htt^{fl/fl}$;UBC-Cre compared with $Htt^{fl/fl}$: 87% reduction, Tukey HSD $P$ = 0.006), but the difference is not significant at the 3 or 9 mo timepoints. This is likely because we had difficulty dissecting

the pancreas without contamination by the surrounding adipose tissue and the high level of proteases in pancreatic tissue causing high variability of HTT measurements compared with other tissues analyzed here (Fig 1C). At 9 mo of age, we euthanized mice and collected pancreases for histological examination. Pancreatic anatomy in all genotypes and treatment groups was normal and contained no diagnostic lesions. Pancreatic acinar cells are filled with zymogen granules (Fig 5A) and featured no signs of degeneration,

**Table 2. Experimental cohort.**

| Genotype | Sex | Treatment | Initiated N (2 mo) | Collected N (14 mo) |
|---|---|---|---|---|
| $Htt^{+/+}$ | M | Vehicle | 10 | 10 |
| $Htt^{+/+}$ | F | Vehicle | 10 | 10 |
| $Htt^{fl/fl}$ | M | Vehicle | 10 | 10 |
| $Htt^{fl/fl}$ | F | Vehicle | 10 | 10 |
| $Htt^{fl/fl}$;CAG-cre | M | Vehicle | 10 | 10 |
| $Htt^{fl/fl}$;CAG-cre | F | Vehicle | 10 | 8 |
| $Htt^{fl/fl}$;UBC-cre | M | Vehicle | 10 | 10 |
| $Htt^{fl/fl}$;UBC-cre | F | Vehicle | 10 | 10 |
| $Htt^{+/+}$ | M | Tamoxifen | 10 | 10 |
| $Htt^{+/+}$ | F | Tamoxifen | 10 | 10 |
| $Htt^{fl/fl}$ | M | Tamoxifen | 10 | 10 |
| $Htt^{fl/fl}$ | F | Tamoxifen | 10 | 10 |
| $Htt^{fl/fl}$;CAG-cre | M | Tamoxifen | 10 | 9 |
| $Htt^{fl/fl}$;CAG-cre | F | Tamoxifen | 10 | 5 |
| $Htt^{fl/fl}$;UBC-cre | M | Tamoxifen | 10 | 10 |
| $Htt^{fl/fl}$;UBC-cre | F | Tamoxifen | 10 | 10 |

**Table 3. Interim takedown cohort.**

| Genotype | Sex | Treatment | Collection age (mo) | | | |
|---|---|---|---|---|---|---|
| | | | 3 | 6 | 9 | |
| $Htt^{fl/fl}$ | M | Tamoxifen | 5 | 5 | 5 | |
| $Htt^{fl/fl}$ | F | Tamoxifen | 5 | 5 | 5 | |
| $Htt^{fl/fl}$;CAG-cre | M | Tamoxifen | 5 | 5 | 5 | |
| $Htt^{fl/fl}$;CAG-cre | F | Tamoxifen | 5 | 5 | 5 | N |
| $Htt^{fl/fl}$;UBC-cre | M | Tamoxifen | 5 | 5 | 5 | |
| $Htt^{fl/fl}$;UBC-cre | F | Tamoxifen | 5 | 5 | 5 | |

necrosis, or inflammation, whereas islet cells were within normal limits. This provides more evidence that early HTT loss is not associated with symptomatic pancreatitis, pancreatic inflammation, or degeneration of the pancreas.

We monitored body weight longitudinally and observed a complex relationship between sex, aging, and HTT loss, but in general, alterations in the bodyweight that may be associated with HTT loss are very subtle in our cohorts (Fig S4). Consistent with Dietrich et al (2017), we observe loss of testicular mass in both Cre lines after tamoxifen treatment, though this did not reach statistical significance as part of our factorial ANOVA comparing all genotypes and treatments (ANOVA Treatment × Genotype interaction $P$ = 0.65). If we restrict our focus to male mice of both Cre genotypes, the impact of tamoxifen treatment is robustly significant, supporting the validity of this reduction (Fig 5B, Treatment ANOVA main effect $P$ < 0001). We recently reported that constitutive hepatic $Htt$ KO results in blistering of the Glisson's capsule when the liver is perfused with PBS (21). We tested this here in a subset of mice at 14-mo and observed the same blistering in 100% of cKO mice tested

(5/5 tamoxifen-treated $Htt^{fl/fl}$;UBC-Cre, 0/5 of $Htt^{fl/fl}$, Fisher's exact $P$ = 0.0079), suggesting this loss of adhesion is replicated after adult-onset HTT loss.

### Central pathology

Consistent with Dietrich et al (2017), we observe large subcortical lesions surrounding accumulations of calcified deposits in Cre mice treated with tamoxifen. In a sagittal section stained with a calcium-sensitive dye, alizarin red, a very clearly-demarcated thalamic lesion becomes clear (Figs 6A and S5A). Manual alignment of each mouse's lesions to the Allen Institute Reference Atlas (23) reveals a relatively circumscribed localization to the posterior complex, ventral posteromedial nucleus, and ventral anterior–lateral complex of the thalamus, which comprise the sensory nuclei of the thalamus, and less frequent lesions found in the mediodorsal, paracentral, and ventral posterolateral nuclei of the thalamus (Figs 6B and S5B and C). Calcifications are also clearly visible in sections stained with cresyl violet (Fig S6). To measure the atomic composition of the lesions more objectively, we turned to scanning electron microscopy-energy-dispersive X-ray spectroscopy (SEM-EDS), which can determine the elemental composition of a sample by characterizing the X-rays emitted as the electron beam sweeps across the sample (24). Paired comparison of spots within and outside the lesion reveals a clear shift—the signal from non-deposit tissue is dominated by carbon, with smaller peaks of calcium, oxygen, and minor peaks for other elements (including silicon, from the underlying glass slide; Fig 6C). This profile is markedly different within the lesions, which are depleted of carbon, with a strong increase in the calcium and phosphate peaks. We do not observe any significant peaks at 6.4 keV, the characteristic energy level of iron, suggesting that the deposits found in our HTT-depleted mice

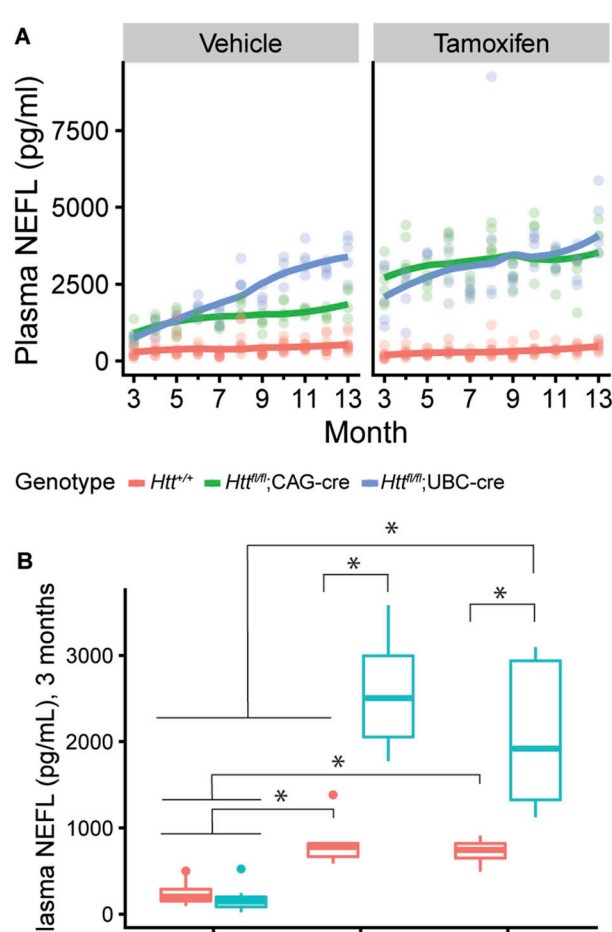

**Figure 2. Plasma levels of neurofilament light chain (NEFL) are increased after HTT loss.**
**(A)** Plasma NEFL levels are increased in all $Htt^{fl/fl}$;CAG-Cre and $Htt^{fl/fl}$;UBC-Cre mice, with the most significant increases being in the tamoxifen-treated groups. Plasma was collected once per month at 3–13 mo of age and measured by chemiluminescent ELISA. **(B)** A cross-sectional view of NEFL levels in each group at 3 mo of age highlights the large increase in NEFL after 1 mo of HTT loss (* indicates Tukey HSD $P < 0.001$, full list of statistical comparisons available in Supplemental Data 1). n = 5 for cre groups, n = 10 for non-Cre groups ($Htt^{+/+}$ and $Htt^{fl/fl}$ combined).
Source data are available for this figure.

are not notably enriched in iron above the lower limit of detection with this technique. Because iron regulatory proteins were found to be dysregulated in Dietrich et al (2017) and the noted role of HTT in iron homeostasis (25), we conducted western blots for transferrin receptor (TFRC) levels. TFRC is a key iron import protein and is up-regulated when intracellular iron levels are low (26). We found that TFRC was significantly increased in tamoxifen-treated $Htt^{fl/fl}$;CAG-Cre mice compared with tamoxifen-treated $Htt^{fl/fl}$ mice (Fig 6D, Welch's T test $P < 0.002$).

We collected sagittal sections of all mice in our study for the examination of histological outcomes. Sections were stained with antibodies for neurons (NeuN, a pan-neuronal marker) or

astrocytes (GFAP) and microglia (IBA1). The regions of the thalamus that displayed calcified deposits were notably devoid of NeuN positive cells, demonstrating a frank loss of neurons where they would otherwise be expected, whereas thalamic areas outside of these regions were not detectably altered (Fig 7A). In addition, thalamic GFAP intensity is significantly higher in tamoxifen-treated $Htt^{fl/fl}$;CAG-Cre mice than in vehicle-treated or wildtype mice (Fig 7B, Tukey HSD $P = 0.0001$), supporting findings by Dietrich et al (2017). IBA1 positive cells appear more numerous and surround the calcified lesions in tamoxifen-treated $Htt^{fl/fl}$;CAG-Cre sections, though quantified IBA1 positive area was not significantly higher overall (ANOVA Treatment × Genotype interaction $P = 0.057$). Whereas we observe increased GFAP and IBA1 intensity in the thalamus, we do not see widespread up-regulation in other regions noted by Dietrich et al, for instance cerebellar GFAP and IBA1 are not up-regulated (Fig S7; GFAP; ANOVA Treatment × Genotype interaction $P = 0.4$; IBA1; ANOVA Treatment × Genotype interaction $P = 0.8$).

## Discussion

This study was conducted to replicate an apparently discrepant set of results between two published studies of the impact of global adult-onset $Htt$ KO. We find that our results largely support those of Dietrich et al (2017) (16) and further reveal significant discrepancies with Wang et al (2016) (15). Whereas we do not fully understand the origin of these differences, some possible explanations are discussed below. In short, we confirm that the chronic, near-complete cellular loss of HTT at 2 mo of age is initially benign but associated with a progressive neurodegenerative phenotype accompanied by increased levels of NEFL in the plasma. We also describe the results of SEM-EDS work confirming that the primarily thalamic lesions are calcium phosphate rich. This constellation of signs raises cautions about a lower limit of HTT silencing that is tolerated in vivo, particularly in the brain. Future studies with more graded HTT lowering are justified to determine whether less extreme lowering carries any of the same risks.

The impact and tolerability of HTT loss in adulthood has been a controversial and complex issue for some time (see recent reviews on the issue (27, 28)). It is now clear that complete loss of HTT, or expression of hypomorphic alleles, is not tolerated during development in mice (2, 3, 4, 29) or humans (5, 6). However, graded reductions of HTT mediated by siRNA approaches in both mice (30) and non-human primates for as long as 6 mo have been reported to be benign (31, 32, 33, 34, 35). Similarly, HD patients treated with tominersen and other CNS-directed HTT-lowering drugs have not been reported to develop subcortical calcification. However, many patients treated with 120 mg doses of tominersen, which resulted in robust lowering of HTT in the CSF, did show a puzzling, transient increase in NEFL levels—the opposite of what was predicted at study onset (12). It may be that the increased CSF NEFL in HD patients treated with tominersen shares a mechanistic origin with the large and sustained increases in plasma NEFL we observe after near-complete HTT loss in mice, though unlike the human trial we

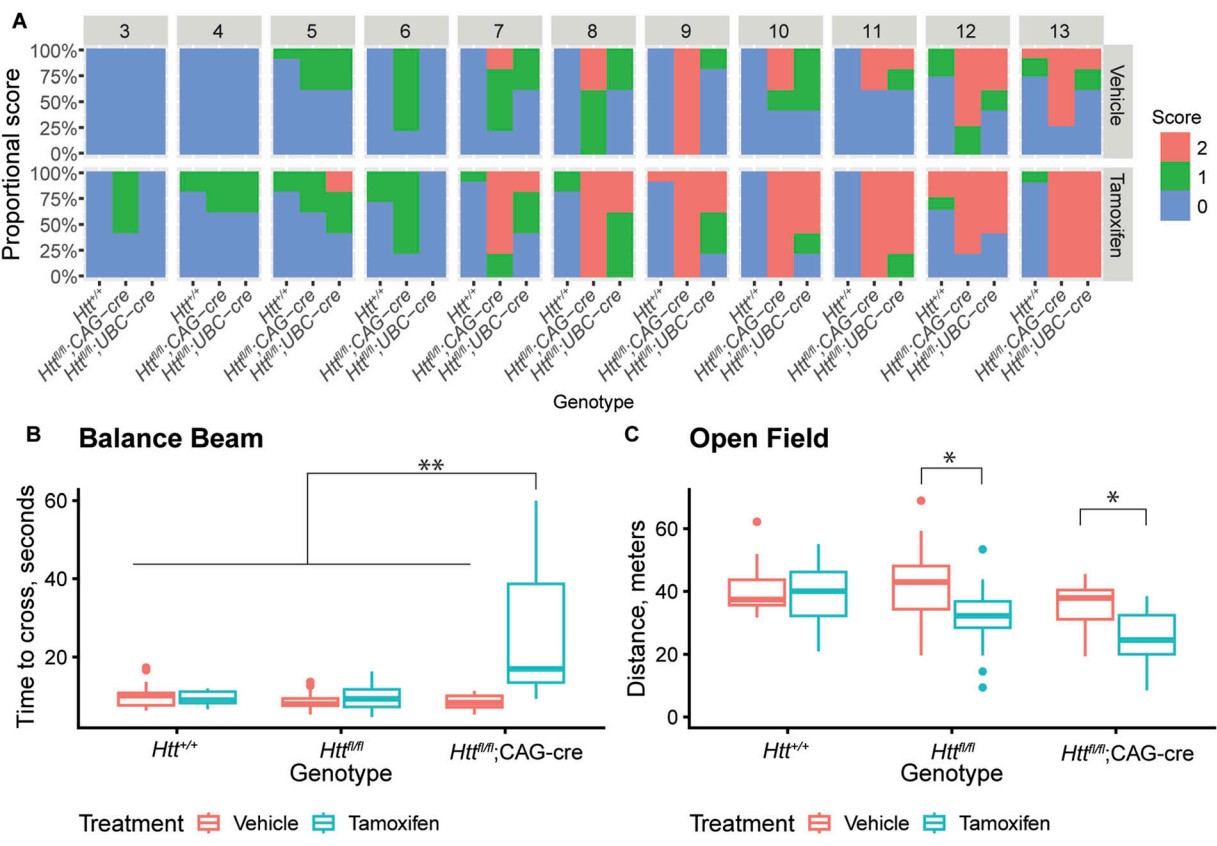

**Figure 3. Progressive behavioral alterations develop after HTT loss.**
**(A)** Tremor incidence and severity in tamoxifen treated $Htt^{fl/fl}$;CAG-Cre and $Htt^{fl/fl}$;UBC-Cre mice progressively worsens with age. n = 5 for cre groups, n = 10 for non-cre groups ($Htt^{+/+}$ and $Htt^{fl/fl}$ combined). **(B)**. At 11 mo of age, tamoxifen-treated $Htt^{fl/fl}$;CAG-Cre mice take significantly longer to cross a balance beam all other groups (** indicates Tukey HSD $P < 0.0001$). N = 15–20/group. **(C)** At 11 mo of age, tamoxifen-treated $Htt^{fl/fl}$ and $Htt^{fl/fl}$;CAG-Cre mice traveled less distance than vehicle treated mice in an open field test, a result of possible tamoxifen toxicity. N = 15-20/group (* indicates Tukey $P < 0.05$, a full list of statistical comparisons available in Supplemental Data 1).
Source data are available for this figure.

did not observe a return to baseline in NEFL throughout the 12 mo of our study.

A major motivation for this study was to clarify whether complete HTT loss is safe, in hopes of informing the rapidly moving field of HTT lowering in HD clinical trials (8). We find that near-complete loss of HTT is poorly tolerated in the brain, suggesting that a straightforward genome editing approach with complete elimination of both *HTT* alleles is likely not a desirable approach to HD therapeutics (36) and that allele-selective genome editing approaches that have been proposed would be better tolerated (37, 38, 39). However, an important limitation of this study is that we used an experimental tool—tamoxifen-inducible Cre—that drives HTT cellular expression from normal levels to near-zero very quickly, which is distinct from pharmacological approaches such as ASOs which result in graded lowering of target genes across cell types (40). The ubiquitous and strong promoters used in this study also drive expression across every cell type of the body (including brain), and so deconvolving the impact of HTT loss on different cell types is not feasible with our approach. Developmental deletion of HTT in specific cell lineages has been shown to be deleterious via distinct

mechanisms—e.g., HTT loss in postnatal neurons and testes results in progressive neurodegeneration and impaired spermatogenesis (41); and loss of *Htt* in *Wnt1*-lineage cells is associated with congenital hydrocephalus (42).

Given that this study was conducted primarily as a replication study, we need to consider points of coherence and points of discrepancy between the now three studies conducted to address the safety of global inducible *Htt* loss. The mice Dietrich et al (2017) relied on were $Htt^{fl/-}$;CAG-Cre (referred in that study as "cKO"), meaning that they developed with 50% of normal total HTT levels before being treated with tamoxifen to drive HTT from 50% towards 0%. In general, we observe very few differences from the findings of this study, apart from body weight, which they observe is consistently reduced in tamoxifen treated "cKO" mice, whereas we observe a more complex relationship between age, genotypes, and sex (Fig S2). This may be because of the confound of different total HTT levels during development in their study (50%) and ours (100%). Interestingly, Dietrich et al (2017) report some unexpected phenotypes in cKO mice not treated with tamoxifen, namely: earlier death, lower body weight (consistent with a previous finding (14)) and impaired motor coordination compared with wildtype mice.

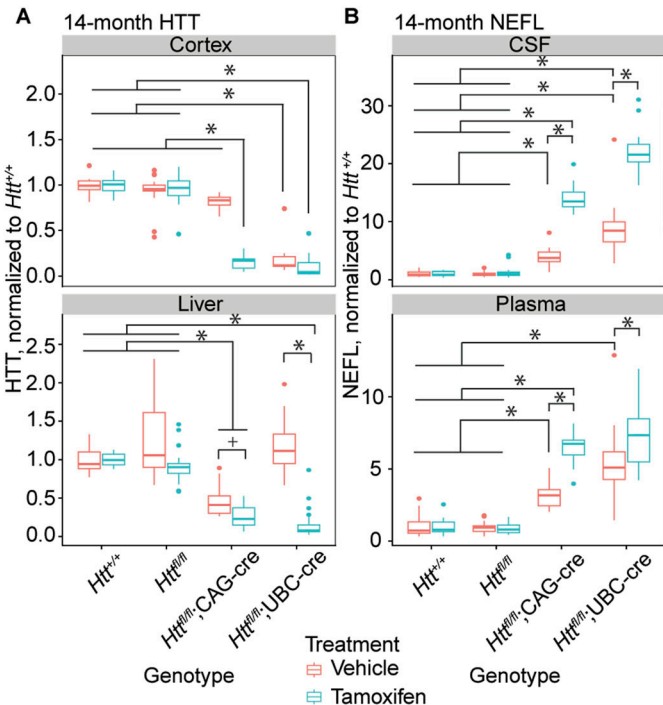

**Figure 4. Terminal HTT levels are lowered and NEFL levels are elevated in tamoxifen-treated cKO mice.**
**(A)** In cortex (above), vehicle-treated *Htt*$^{fl/fl}$;CAG-Cre mice have HTT levels similar to control groups. However, vehicle-treated *Htt*$^{fl/fl}$;UBC-cre have reduced HTT levels similar to tamoxifen-treated mice, demonstrating leakiness of UBC-cre in the CNS. In liver, the inverse can be seen, demonstrating leakiness of CAG-cre in the periphery. In both tissues, tamoxifen-treated cKO mice show robust reduction in HTT. N = 12–20/group. **(A, B)** NEFL levels from CSF (above) and plasma (below) at 14 mo of age show significant increases which correspond to the reduction in HTT seen in panel (A). N = 11–20/group (* indicates Tukey HSD *P* < 0.01, + indicates Tukey HSD *P* < 0.05, a full list of statistical comparisons available in Supplemental Data 1).
Source data are available for this figure.

This suggests that 50% HTT expression is associated with significant important age-related phenotypes, which were likely exacerbated by additional loss of HTT expression that we observe occurs with CAG-cre and the *Htt* flox allele.

Our findings are more discrepant with those of Wang et al (2016), who observed acute pancreatitis after treating *Htt*$^{fl/fl}$;CAG-Cre at 2 mo of age with tamoxifen, but no deleterious motor or pathological phenotypes with mice aged to 18 mo of age after knocking out *Htt* at 4 or 8 mo of age. We speculate that this apparent discrepancy may arise from selective toxicity of tamoxifen, a compound with a very narrow therapeutic index in both humans and mouse models (43), particularly in 2-mo-old animals. The levels of tamoxifen injected by the Wang et al (2016) were 100 mg/kg for five consecutive days, whereas we and Dietrich et al (2017) used 75–77 mg/kg. Whereas the higher tamoxifen levels used by Wang et al (2016) may explain the early toxicity observed, it remains unclear why *Htt*$^{fl/fl}$;CAG-Cre mice would be selectively vulnerable to this compared with non-floxed control mice, who survived. The apparently benign mouse phenotype observed by Wang et al (2016) after knocking out *Htt* at 4 and 8 mo of age may reflect the reduced amount of time those mice experienced acute

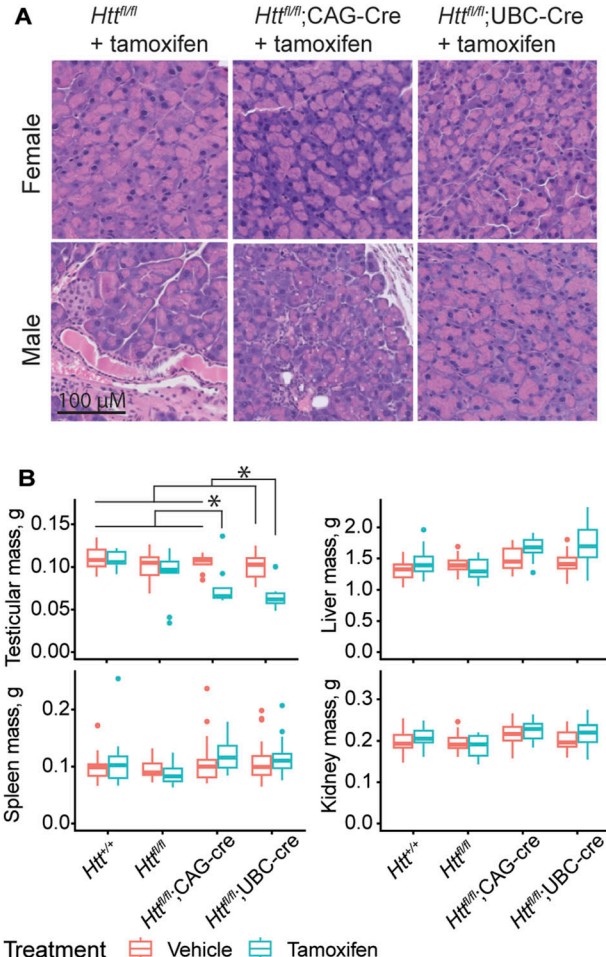

**Figure 5. Pancreatic histology is normal after early and prolonged HTT loss.**
**(A)** H&E staining in the pancreas shows no gross abnormalities of in any genotype of tamoxifen treated mice, with acinar cells visibly full of zymogen granules. **(B)** Testicular weights are lower in both tamoxifen-treated Cre lines compared with all other groups lines. Other organ weights show no significant changes (* indicates Tukey HSD *P* < 0.05, a full list of statistical comparisons available in Supplemental Data 1). N = 12–25/group.
Source data are available for this figure.

HTT loss before euthanasia, though we observe clear behavioral outcomes such as tremors as early as 5 mo post-HTT loss (Fig 3A). It may be the case that silencing HTT at 2 mo of age is particularly deleterious because of the developmental stage at which silencing occurs.

We are intrigued that complete HTT loss reproducibly causes subcortical calcification, a very distinctive neuropathological lesion. Many human conditions, and some of their animal models, have similar lesions. One such family of diseases—type I interferonopathies—is caused by hyperactivation of type I interferon responses secondary to deficits in nucleic acid homeostasis (44, 45). Humans with these conditions often present with subcortical calcification, as do many of the mouse models generated to study them. Similarly, patients with Cockayne syndrome—which arises from deficient DNA repair—often present with subcortical calcification (46). In Down syndrome,

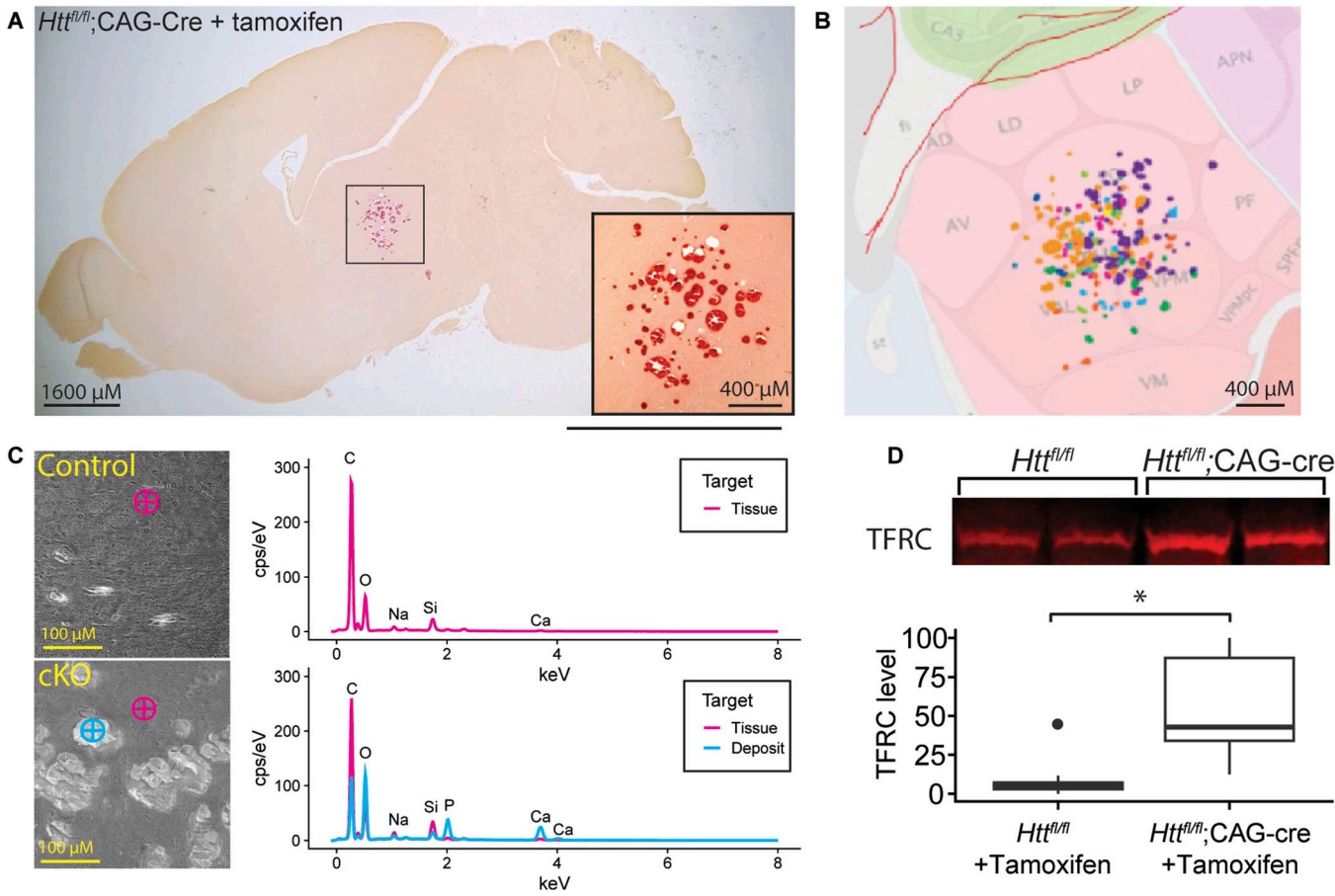

**Figure 6. Histological analysis reveals overt thalamic calcification and inflammation after HTT loss.**
**(A)** Large subcortical deposits are strongly positive for calcium marker alizarin red in a cKO mouse at 14 mo of age. **(B)** Overlay of Allen Brain Atlas on lesions demonstrates the average location of the lesions, where lesions from every mouse are represented by a different color (see Fig S4 for more detail). **(C)** (Left) SEM-EDS images reveal that deposits are absent from control tissue but present in cKO tissues. (Right) EDS spectra reveals that deposits show enrichment for oxygen (O), and phosphate (P), and calcium (Ca), suggesting these are likely $Ca_2PO_4$ deposits. **(D)** Western blot of cortical lysates reveals increased levels of TFRC in cKO mice; transferrin receptor (TFRC) signal was normalized to total protein per well (* indicates Tukey HSD $P < 0.01$).
Source data are available for this figure.

calcification of the basal ganglia has been widely observed and has been attributed to enhanced brain aging (47, 48). Another potentially significant point of comparison is with familial idiopathic basal ganglia calcification (Fahr's disease) which is a common cause of intracranial calcification in humans (49) An emerging picture of the pathology in familial idiopathic basal ganglia calcification is that it arises from deficiencies in ion homeostasis across the blood–brain barrier, either via mutations in ion transporters themselves (*SLC20A2* (50) or *XPR1* (51)), or in genes that play a role in pericyte growth factor responses (*PDGFB* (52) or *PDGFRB* (53)) and thereby blood–brain barrier permeability. In recently proposed models, the high local calcium and phosphate concentrations drive deposition of $Ca_2PO_4$ deposits near tissues which produce CSF, which drives the anatomic localization of these highly localized deposits (49). Future detailed work comparing the phenotypes of mice with acute loss of *Htt* and these other syndromes associated with subcortical calcification is warranted by the reproducibility and specificity of this phenotype.

# Materials and Methods

### Mice

We generated two lines of conditional *Htt* KO mice (cKO) by crossing CAG-Cre (JAX stock 004682) or UBC-Cre (JAX stock 007001) to a *Htt*$^{fl}$ mouse line (4). All mouse strains were on a C57Bl/6 background, including *Htt*$^{fl}$, which was previously backcrossed and confirmed for congenicity (21). Mice were maintained at homozygosity for the *Htt*$^{fl}$ allele and heterozygosity for the Cre allele. All mice were bred at the McLaughlin Research Institute (MRI) and housed in cages of 3–5 mice with access to food and water ad libitum unless otherwise mentioned. Vivarium lights were on a 12-h light/dark cycle. The MRI institutional animal care and use committee approved all procedures under protocol 2020-JC-29. Mice were genotyped with the following primers: for Htt flox PGK Reverse 5′ CTAAAGCGCATGCTCCAGACTG-3′, Flank Forward 5′-AGATCTCTGAGTTATAGGTCAGC-3′, and Flank Reverse 5′-CATTTGATTCTTACAGGTAGCCTG-3′ (band sizes of 320 bp for the Floxed PGK Reverse and Flank Forward and 180 bp for the WT with

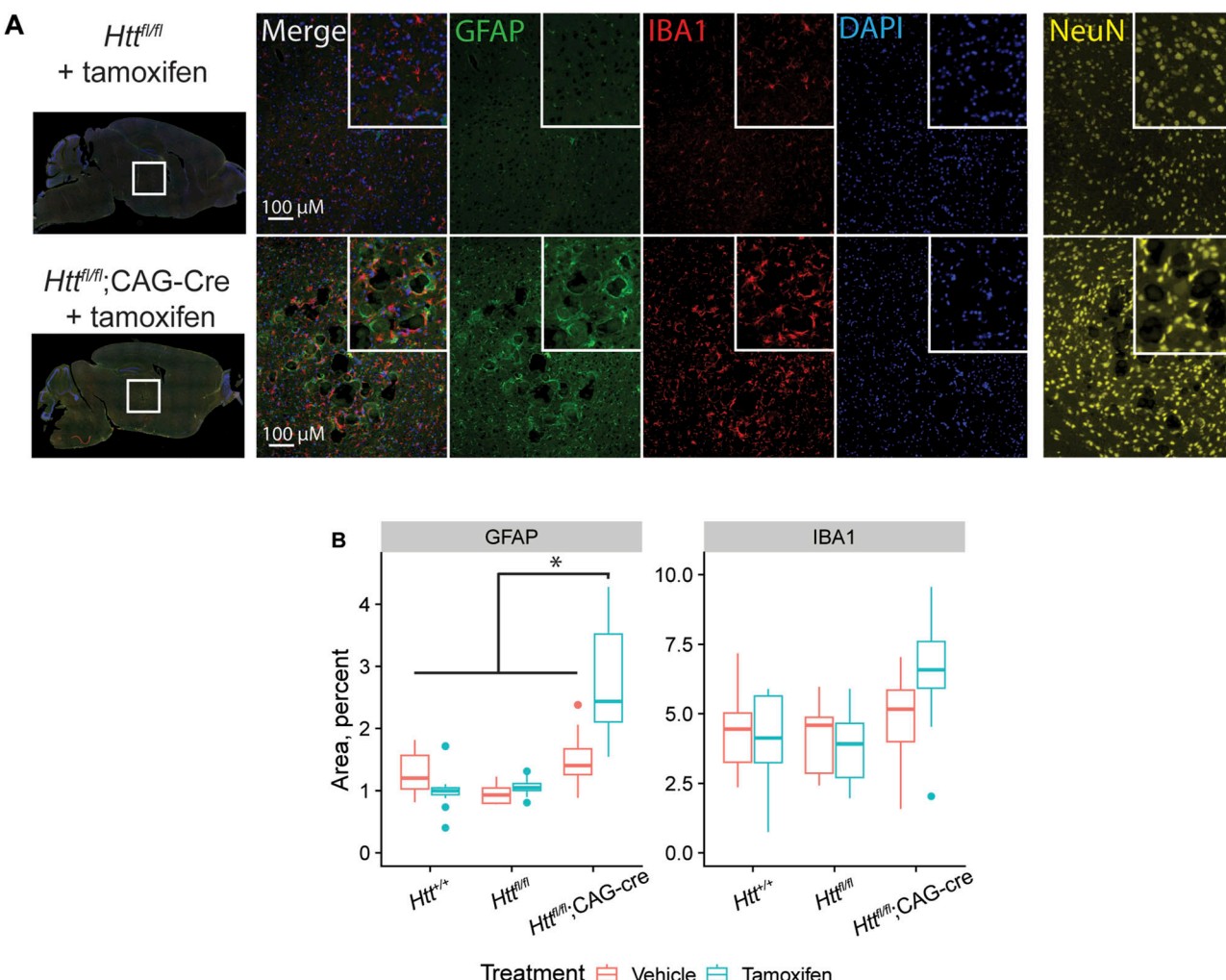

**Figure 7. Immunoreactivity for glial markers IBA1 and GFAP is increased after HTT loss.**
**(A)** Immunohistochemical staining for microglia (IBA1, red) and astrocytes (GFAP, green), nuclei (DAPI, blue) reveals inflammation in the thalamus surrounding the calcified deposits. Additional staining on an equivalent brain section for neurons (NeuN, yellow) shows a clear loss of neuronal cells in regions with calcified deposits. Inset images are 2× magnification of full size images. **(B)** Quantification of IBA1 and GFAP area reveals significant increase in GFAP area in cKO mice, whereas IBA1, which qualitatively appear increased surrounding the thalamic lesions, is not significantly increased in cKO thalamus overall (* indicates Tukey HSD $P < 0.01$, a full list of statistical comparisons available in Supplemental Data 1).
Source data are available for this figure.

Flank Forward and Flank Reverse), for UBC-cre 25285 Forward 5′-GACGTCACCCGTTCTGTTG-3′, oIMR9074 Reverse 5′-AGGCAAATTTTGGTG-TACGG-3′ (band size 475 bp), and for CAG-cre oIMR5984 Forward 5′-GCTAACCATGTTCATGCCTTC-3′, oIMR9074 5′-AGGCAAATTTTGGTGTACGG-3′ (band size of 180 bp). Band sizes were determined on a 1.5% agarose gel with 0.5% TBE buffer. Tamoxifen (T5648; Sigma-Aldrich) was administered via I.P. injection at 75 mg/kg once daily for 5 consecutive days at 2-mo of age. Tamoxifen was reconstituted in pharmaceutical grade corn oil (C8267; Sigma-Aldrich) before administration.

## Recombination analysis

DNA recombination was tested using semi-quantitative PCR (Fig S6). Genomic DNA was extracted from cortex and liver using the IBI gMax Mini Kit (IB47281), using the manufacturer's protocol for solid tissue. PCR was performed using the pgk, Hdhpr13, and Hdhrec9

primers flanking the deletion site as described by reference 16, resulting in two bands, a 220 bp one showing the amount of recombined allele and a 210 bp one showing the amount of unrecombined allele. PCR products were run on 2.2% agarose gels (57031; Lonza) and imaged using Philips VLounge software. Band intensity was quantified in ImageStudio Lite software. Recombination efficiency was calculated by dividing the intensity of the recombined band by the total signal of the recombined plus unrecombined bands, so that a higher value is equivalent to more recombination.

## Behavioral analysis

Mice were assessed monthly beginning at 3 mo until 13 mo of as part of a modified SmithKline Beecham, Harwell, Imperial College, Royal London Hospital Phenotype Assessment (SHIRPA) (16, 19).

Briefly, mice were assessed for 13 measures (Table S1) where normal behavior received a score of "0" and severity of abnormalities were indicated with higher scores ranging from 1–3, depending on the measure as outlined on Table S1. To assess thigmotaxis behavior, mice were recorded for 10-min in an open field box (40 × 40 cm) and activity was quantified with ANY-Maze software (Stoelting) for measures including distance traveled and time spent within 6.5 cm of walls. The next two days, mice were trained to cross a 1-m long elevated balance beam and traversal time was measured on the third day as described (54). For training, mice were placed on one end of the beam, which had a 60-W lamp, and encouraged to cross the beam towards the opposite side, which had a dark box with home-cage nesting material. Mice were trained with three trials on a 12-mm wide beam with 15 s intervals between trials. Mice were then given a 10-min break and this was repeated on a 6-mm wide beam. After 2 d of training, mice were similarly placed at the start of the 6-mm beam and the time to traverse was measured and averaged for three trials. Any time over 60 s was scored as 60 s, any falls were scored as 60 s.

### Protein quantification (HTT, NEFL, Tfrc)

Total HTT levels were quantified using an electro-chemiluminescent ELISA measured with a MESO QuickPlex SQ120MM (Meso Scale Discovery; MSD) according to previously described methods (55). Tissues were homogenized in non-denaturing lysis buffer in tubes containing 1.4 mm zirconium oxide beads at 6 m/s in three 30 s intervals with 5 min on ice between rounds. Lysates were centrifuged for 20 min at 20,000$g$ at 4°C. Supernatant was transferred to a new tube and centrifuged again for 20 min at 20,000$g$ at 4°C, then transferred to a second new tube. 96-well MSD plates (MSD, L15XA-6) were coated with capture antibody (CHDI-90002133, 8 $\mu$g/ml) in carbonate-bicarbonate coating buffer for 1 h with shaking at 750 RPM on a Titramax 1000 shaker (Heidolph). Plates were then washed three times with wash solution (0.2% Tween-20 in PBS) and blocked with blocking buffer (2% BSA, 0.2% Tween-20 in PBS) for 1 h with shaking, then washed three more times. Samples were diluted (brain: 2 $\mu$g/$\mu$l, liver: 4 $\mu$g/$\mu$l) in 20% MSD lysis buffer, 80% blocking buffer and incubated for 1 h with shaking. Plates were washed three more times, followed by incubation with sulfo-tag conjugated secondary antibody (D7F7-Sulfo-Tag, 1:1,800) for 1 h with shaking. Plates were washed a final three times, and Read Buffer B (MSD, R60AM-4) was added to the plate before reading on a QuickPlex SQ 120 MM.

Neurofilament light levels were quantified using the Neurofilament L Assay (MSD, K1517XR-2) according to manufacturer directions. Briefly, plates were coated with biotinylated capture antibody at 1:16.5 in diluent 100 and incubated for 1 h with shaking at 700 RPM on a Titramax 1000 shaker (Heidolph). They were then washed 3 times with wash buffer, followed by addition of the samples diluted at 1:10 in diluent 12, and incubated for 1 h at 700 RPM. After three more wash steps, secondary antibody was added at 1:100 in diluent 11 and incubated for 1 h at 700 RPM, followed by a final three washes. Read Buffer B was added, and the plate was read on a QuickPlex SQ 120 MM.

TFRC was quantified by Western blot. Samples were run on a 10% Bis-tris gel (NO0301; Thermo Fisher Scientific) for 40 min at 200 V.

The iBlot 2 system with a PVDF membrane was used for protein transfer. Membranes were blocked with Intercept blocking buffer (927-60001; Licor) for 1 h, and incubated with TFRC primary antibody (13-6800; Thermo Fisher Scientific) for 1 h, followed by washing and then secondary antibody (926-68070; Licor) for 1 h. Total protein per lane was quantified using Revert total protein stain (Li-Cor) and each lane was normalized to the total protein per lane. Blots were imaged on a Li-Cor Odyssey CLx and quantified using ImageStudio (Li-Cor).

### Plasma chemistry

Clinical chemistry plasma analysis for 11 analytes (Fig S3) was completed using an Atellica Clinical Chemistry Analyzer (Siemens) at Phoenix Central Laboratories. Plasma was collected via cardiac puncture following a lethal injection of sodium pentobarbital (Fatal Plus, Henry Schein) at 14 mo (SD = 5.7 d). Plasma was flash frozen following collection in heparinized microtainers (cat. no. 365965; BD) and purification by centrifugation at 1,300$g$ for 10 min, followed by 2,500$g$ for 15 min.

### Histology

Hemibrains were drop-fixed in 10% neutral buffered formalin (BBC biochemical) for 48 h. After fixation, they were stored in PBS + 0.02% sodium azide until they were paraffin embedded, cut into 5-$\mu$m sections, and mounted on glass slides. For fluorescent immunohistochemistry, sections were deparaffinized and rehydrated to distilled water as described previously (21). Sections underwent heat-mediated antigen retrieval in a water bath for 20 min in pH = 9 Tris–EDTA buffer. After washing with water, slides were blocked for 1 h with 20% goat serum in PBS. Primary antibodies were incubated at the following concentrations in PBS with 20% goat serum, 1% BSA, and 0.1% Tween20: GFAP (MAB3402, 1:500; EMD), Iba1 (019-19741, 1:500; Wako), NeuN (ABN78, 1:750; EMD). Samples were washed with PBS three times and incubated with the appropriate secondary antibody (Alexa Fluor) diluted 1:1,000 for 1 h at room temperature in antibody diluent. Samples were washed with PBS three times and coverslips were applied with Vectashield DAPI hardset mounting medium (H-1500-10; Vector labs). Slides were imaged on a Leica DMI600 widefield microscope. Quantification was completed in FIJI/ImageJ by thresholding to select image areas positive for immunostaining.

For calcium staining, sections were deparaffinized and rehydrated to distilled water as described previously. Sections were stained with a 2% alizarin red solution (C.I. 58005) with pH adjusted to 4.2 using ammonium hydroxide, then rinsed in water. They were dehydrated with 20 dips in 100% acetone, then 20 dips of 1:1 acetone/xylene and cleared in 100% xylenes. Coverslips were applied with Permount (Fisher Chemical SP15). Slides were imaged on an Olympus BX51. For pancreatic pathology, H&E-stained pancreatic sections were scored by a veterinary pathologist (Zoetis Reference Laboratories) blinded to genotype.

## SEM-EDS

Brains from 12-mo-old mouse were drop-fixed in 10% neutral buffered formalin (BBC biochemical) for 48 h followed by cry-oprotection by sinking in 30% sucrose for 48 h and subsequently flash frozen in isopentane on dry ice. Brains were embedded in cutting medium (OCT) and cut in 20 $\mu$m sections on a cryostat. Brains were mounted onto slides and allowed to dry overnight and coated in Au:Pd before imaging. Slides were imaged on a FEI Sirion XL30 Scanning Electron Microscope (SEM) equipped with an Oxford Instruments Energy Dispersive X-ray Spectrometer (EDS) under vacuum conditions. EDS spectra were collected from lesion areas and adjacent non-lesioned tissue or equivalent region of non-lesioned brains.

## Statistical analysis

Primary data for our analyses were processed using R (56). For factorial tests, we used ANOVA with Tukey's tests for post hoc analyses where appropriate. Data presented in Figs 1, 2, 3, 4, 5, 6, and 7, and S1, S2, S5, S6, and S7 used boxplots—horizontal lines indicate 25th, 50th, and 75th percentile, whereas the vertical whiskers indicate the range of data. Data falling outside 1.5 times the interquartile range are graphed as isolated points but were not excluded from statistical analysis. All statistical comparisons are available in Supplemental Data 1. Graphics were produced using ggplot2 (57), Illustrator (Adobe), and BioRender.com.

# Data Availability

All raw data underlying these analyses, including tabular molecular and behavioral data, R scripts and microscopy images are available as a stable digital repository at Dryad at with the persistent reference 58.

# Supplementary Information

# Acknowledgements

Funding for this project was provided to JB Carroll via an unrestricted research grant from Ionis Pharmaceuticals and a research agreement between CHDI Foundation and UW and WWU (Record number # A-18222). The authors thank Holly Kordasiewicz for thoughtful discussions around the design of this study. The authors thank Scott Braswell for assistance with SEM-EDS at the University of Washington Molecular Analysis Facility, a National Nanotechnology Coordinated Infrastructure (NNCI) site which has partial support from the National Science Foundation via awards NNCI-1542101 and NNCI-2025489. The authors thank the staff at McLaughlin Research Institute that assisted with behavioral assays and tissue collection including Serena McElroy, Megan Ratz-Mitchem, June Pounder, Kaela Davey with partial support from NIH COBRE award 1P20GM152335-01.

## Author Contributions

RM Bragg: conceptualization, data curation, formal analysis, validation, investigation, visualization, methodology, project administration, and writing—original draft, review, and editing.
EW Mathews: data curation, formal analysis, validation, visualization, methodology, and writing—original draft, review, and editing.
A Grindeland: investigation, methodology, and writing—review and editing.
JP Cantle: conceptualization, investigation, methodology, and writing—review and editing.
D Howland: conceptualization, resources, and writing—review and editing.
T Vogt: conceptualization, resources, and writing—review and editing.
JB Carroll: conceptualization, supervision, funding acquisition, and writing—original draft, review, and editing.

## Conflict of Interest Statement

D Howland and T Vogt are full-time employees of CHDI Foundation. JB Carroll is a paid advisor for Cajal Neuroscience and Guidepoint. RM Bragg received consulting fees from Takeda. All other authors have no conflict of interest to report.

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
