## [Reviewer comments · Life Science Alliance]

Life Science Alliance

Global Huntingtin Knockout in Adult Mice Leads to Fatal Neurodegeneration that Spares the Pancreas

Robert Bragg, Ella Mathews, Andrea Grindeland, Jeffrey Cantle, David Howland, Thomas Vogt, and Jeffrey Carroll
DOI: <https://doi.org/10.26508/lsa.202402571>

Corresponding author(s): Jeffrey Carroll, University of Washington

Review Timeline:

Submission Date:	2024-01-04
Editorial Decision:	2024-02-02
Revision Received:	2024-06-13
Editorial Decision:	2024-06-14
Revision Received:	2024-06-21
Accepted:	2024-06-24

Transaction Report:

February 2, 2024

Re: Life Science Alliance manuscript #LSA-2024-02571

Dr. Jeffrey B Carroll
University of Washington
Department of Neurology
HMC #359660
325 9th Ave
Seattle, WA 98104-2499

Dear Dr. Carroll,

Thank you for submitting your manuscript entitled "Global Huntingtin Knockout in Adult Mice Leads to Fatal Neurodegeneration that Sparing the Pancreas" to Life Science Alliance. The manuscript was assessed by expert reviewers, whose comments are appended to this letter. We invite you to submit a revised manuscript addressing the Reviewer comments.

Thank you for this interesting contribution to Life Science Alliance. We are looking forward to receiving your revised manuscript.

Sincerely,

B. MANUSCRIPT ORGANIZATION AND FORMATTING:

Reviewer #1 (Comments to the Authors (Required)):

Global huntingtin knockdown in adult mice

This was a replication study, whereby Huntingtin was knocked down from two months of age in mice. Behavior and histology were used to determine toxicity. By and large the study replicated that of Dietrich at all who found thalamic calcification in the HTT knockdown, but not the pancreatitis that had been seen by Wang et al. Although this was a replication study, and they had opportunity to use the same mice, a different approach was taken from that of Dietrich. A reason for this was not given.

The study seems well conducted, and their comparisons are generally upheld. Some of their data replicate those of Dietrich but do not replicate those of Wang et al.

There are a number of elements of the paper that are not well explained or discussed clearly. These omissions should be addressed. (See specific comments below). In particular, there is a lack of detail regarding the histology. Dietrich et al found severe changes in behavior and histology that the authors of the current paper have not attempted to replicate.

The title describes 'fatal neurodegeneration'. They do not describe neurodegeneration, and as far as I could see they killed the mice because of ulcerative dermatitis. Where is the evidence of fatal neurodegeneration? The title needs to be revised to represent their findings accurately.

They talk about 'progressive neurodegeneration' in their abstract, but there are no data included regarding the loss of neurons in the thalamus or any other region. There is an increase in NFL, but this is not the same as neurodegeneration. They identified areas in which the calcification occurred as being posterior complex, ventral posteromedial nucleus, and ventral anterior-lateral complex of the thalamus. This is a large number of nuclei in the thalamus and would lend themselves to quantification. They need to include data showing that there is neurodegeneration in these regions. A number of questions could be answered with histological examination. For example, is there equivalent neuronal loss from each of these thalamic areas? What about other areas of the thalamus? They should quantify both calcified and non-calcified regions.

It is particularly notable that there is no information about neuronal changes (or lack of changes) in the stratum. Given that HD is generally considered to be a striatal disorder, and that they saw pronounced knockdown in the stratum, it would be useful to include these data- even if they show no change. This would compare directly with the study of Dietrich.

There is little discussion about why Huntington knockdown should affect the thalamus. It is implied that the changes are specific to the thalamus, but they do not show data from other regions, e.g. stratum or cortex, which would make the specificity of the thalamic changes more convincing.

Leakiness of the vectors is a concern, and although the authors have suggested that each model represents the effective knockout of either CNS or peripheral HTT, their interpretation of behavior in the mice in this context is not very convincing.

Since the thalamus was the only region that showed any pathology, leak assessment for this region should be included. It is not clear why this was not included.

Minor comments.

Many of the Figures are not well labelled and of low quality. The authors should review them to make them clearer. For example,

There is no key for symbols in Fig 1A.

X axis titles are needed for many panels in most figures, e.g. Figs 1B, 1C, 2A, 3A, 4A, 5B, 6C, D etc.

Statistics indicated in many panels, but details are not included in the fig legends for many figures.

Figure 5 is supposed to show lack of pancreatic pathology, but most of the illustrated sections show acinar cells, not islets.

Better representative images should be chosen to include both and then labelled clearly. Data from pancreas mass should have been included. They said that the pancreas is difficult to dissect, however many other investigators manage this. In the absence

of mass data or clear histology, to show there was no pancreatitis, they could have measured Serum amylase, serum lipase, pancreas trypsin activity (as done in Wang et al.)

Figure 6 needs labeling.

Fig 7 A needs labeling. Which part of the thalamus is shown? Was there an increase in GFAP throughout the region? It would be useful to see this at a resolution similar to that shown in Fig 6.

Details of mice missing from the methods, e.g. when tamoxifen treatment was started. They describe treatment as being lowering at 2 months -this should really be from 2 months since their first time point was 3 months.

Is the balance beam test validated? If so, provide a reference.

Is not clear why, given this was a replication study, that they did not use the same mice as either of the previous studies. They should explain the rationale for using different breeding strategies.

line 56 there is no need to abbreviate loss of function.

line 57 gnomAD need spelling out in full.

line 70 a reference is needed for lack of effect of 50% HTT expression.

Reviewer #2 (Comments to the Authors (Required)):

The development of huntingtin-lowering therapies has been a major focus in recent years and a wide range of huntingtin lowering treatments are in clinical trials. However, the levels to which the huntingtin protein can be lowered without adverse consequences is unknown and a potential confound for many of these approaches. It has been known for close to 30 years that the complete loss of huntingtin causes severe developmental symptoms in mice and more recent data has shown that this is also true in humans. However, it is the extent to which huntingtin can be lowered in adulthood that is relevant to the design of treatments that will be safe in the clinic. This question has previously been addressed by conditional knockout studies in mice. One study found that the early induction of global HTT loss caused fatal pancreatitis, but that lowering HTT at later ages was benign. Another study found that postnatal HTT loss was associated with subcortical calcification and neurodegeneration but did not report fatal pancreatitis.

In this study, the authors have set out to better understand the risks posed by the widespread inducible loss of HTT. They utilised two inducible tamoxifen Cre lines, each under the control of ubiquitous promoters. This was important as these lines exhibited complementary levels of leakiness; broadly speaking, the UBC promoter was leaky in the brain but not the periphery, whereas the CAG promoter was leaky in the periphery but not the brain. They characterised the pattern of wild-type HTT loss throughout the brain and periphery when conditional knock-out was initiated at 2 months of age. They found that widespread loss of HTT at 2 months of age led to a wide range of phenotypes, including subcortical calcification, but did not result in acute pancreatitis or histological changes in the pancreas. Importantly, they found that HTT loss was accompanied by a robust and sustained increase in the levels of neurofilament light chain (NEFL), a peripherally accessible biomarker that is widely in HD clinical trials. They confirmed that the complete loss of HTT in mice is associated with pronounced risks, including progressive subcortical calcification and neurodegeneration.

In all cases the data strongly support the conclusions of the paper.

This well-conducted and well-powered study is extremely important and will be of great interest to all researchers and clinicians in the Huntington's disease community.

There are some minor comments.

Gene and protein nomenclature for huntingtin should be HTT (*italics*) for human gene, Htt (*italics*) for mouse gene and HTT for the human and mouse protein.

The gene and protein symbols for neurofilament light chain as approved by the HUGO Nomenclature Committee are NEFL (*italics*) and NEFL respectively (see Caron et al. 2022 Brain Comm. Fcac309), although there are many aliases for the protein.

Introduction

Page 4, line 89: typo 'at beginning at'

Results

FigS6. It would be helpful if symbols were defined in the figure legend.

Fig.3C legend. Could the authors add quantification and p values to the statement 'Httfl/fl;CAG-Cre mice travelled slightly less distance than control mice'

Fig 5A the comparative images of pancreatic histology do not appear to have been presented at the same magnification. Could the authors show higher magnification images and point out the aspects of histology mentioned in the text e.g. zymogen granules.

Fig 6. Could the authors define Tfr in the legend?

Materials and Methods

The authors do not mention the mouse strain background that they are using in their studies and whether this is the same background as was used by Dietrich et al. and by Wang et al.

Behavioral analysis - page 12 line 349. Table 4 should be Table S1

Reviewer #3 (Comments to the Authors (Required)):

Huntingtin, the protein affected in Huntington's disease has many well-described functions and the consequences of lowering this protein using various methods including tamoxifen Cre inducible systems has been described. In Huntington's disease (HD), typically one allele has the mutation. Non-allele specific lowering of huntingtin is an approach being evaluated for treatment of HD in mice and in patients. This replication study addresses the effects of lowering huntingtin in C57Bl6 mice using two inducible systems reported by others. In the present work huntingtin was reduced at 2 months and the consequences were examined at 14 months. This well-written and nicely illustrated study presents a comprehensive behavioral, molecular and histological picture that replicates some of the adverse phenotypes reported in the two prior studies. The evidence that lowering huntingtin can signal an elevation in plasma and CSF levels of Neurofilament protein, a marker of compromised axonal integrity, is a significant new contribution. Overall, this work adds considerable value because of the importance of this issue for future clinical therapy, the new data reported, and the care the authors took in detailing the caveats associated with the experimental models.

Scale bars need to be added to some of the figures.

Fig 7 An inset to show cellular detail for GFAP at least would be helpful. Are the dark circular areas cross sectioned blood vessels?

Adding P values or asterisks to box plots for significant comparisons would be very helpful.

Hildich-McGuire et al., 2000 is another good paper to site.

Reviewer #1 (Comments to the Authors (Required)):

We thank the reviewer for their careful reading of our manuscript, which they accurately summarize below. Our responses to each of the concerns they raised are below in bold, interspersed with their original comments.

Global huntingtin knockdown in adult mice

This was a replication study, whereby Huntingtin was knocked down from two months of age in mice. Behavior and histology were used to determine toxicity. By and large the study replicated that of Dietrich at all who found thalamic calcification in the HTT knockdown, but not the pancreatitis that had been seen by Wang et al. Although this was a replication study, and they had opportunity to use the same mice, a different approach was taken from that of Dietrich. A reason for this was not given.

We thank the reviewer for highlighting this - we have noted in the first section on mouse line establishment that we chose not to use Htt hemi-null mice that were used by Dietrich et al, to exclude the hemi-null as a potential contributor to the results. We have updated the text with the following:

“In contrast to Dietrich et al., but in concordance with Wang et al., we maintained full HTT expression though development by using homozygous Htt flox alleles, rather than using a Htt flox allele crossed to Htt null allele, as we did not want to investigate Htt loss of function in the hemi-null context. “

The study seems well conducted, and their comparisons are generally upheld. Some of their data replicate those of Dietrich but do not replicate those of Wang et al.

There are a number of elements of the paper that are not well explained or discussed clearly. These omissions should be addressed. (See specific comments below). In particular, there is a lack of detail regarding the histology. Dietrich et al found severe changes in behavior and histology that the authors of the current paper have not attempted to replicate.

With regards to behavior, we chose to replicate the SHIRPA testing, where we confirmed the presence of severe tremors and some gait abnormalities. We did not detect hindlimb clasping or kyphosis and we have clarified the consistencies and discrepancies between our study and the Dietrich study in the text. To probe this further, Dietrich tested mice on an elevated rotorod - designed to detect relatively subtle changes in motor skills. Rather than use a task with complex apparatus and complex training requirements, we used a simple balance beam test to confirm the motor impairment findings described in Dietrich et al.

The title describes 'fatal neurodegeneration'. They do not describe neurodegeneration, and as far as I could see they killed the mice because of ulcerative dermatitis. Where is the evidence of fatal neurodegeneration? The title needs to be revised to represent their findings accurately.

We show evidence of neurodegeneration based on the loss of thalamic tissue, which is replaced with large calcified deposits, and feel it's justified to say fatal due to severe neurological impairment indicated by tremors that was inhibiting balance and ability to ambulate in the cage.

Based on this, we're comfortable with keeping the title but we defer to the reviewers/editor.

Some potential alternatives to the current title (Global Huntingtin Knockout in Adult Mice Leads to Fatal Neurodegeneration that Spares the Pancreas) are listed here:

Global Huntingtin Knockout in Adult Mice Leads to Neurodegeneration, Spares the Pancreas

Global Huntingtin Knockout in Adult Mice Leads to Thalamic Neurodegeneration, Spares the Pancreas

They talk about 'progressive neurodegeneration' in their abstract, but there are no data included regarding the loss of neurons in the thalamus or any other region. There is an increase in NFL, but this is not the same as neurodegeneration. They identified areas in which the calcification occurred as being posterior complex, ventral posteromedial nucleus, and ventral anterior-lateral complex of the thalamus. This is a large number of nuclei in the thalamus and would lend themselves to quantification. They need to include data showing that there is neurodegeneration in these regions. A number of questions could be answered with histological examination. For example, is there equivalent neuronal loss from each of these thalamic areas? What about other areas of the thalamus? They should quantify both calcified and non-calcified regions.

We thank the reviewer for identifying this weakness. As noted in the text, we observed calcified deposits spread across several thalamic areas, centered around the posterior complex. The number and precise location of deposits was slightly variable between mice and we suspect identifying a precise initiation point at this age is not possible when calcifications have had time to form in such large abundance. In response to this comment, we attempted to quantify the loss of neurons in precise thalamic locations but found this difficult with our remaining samples. We plan to follow up on this phenomenon with a dense timecourse of mice to identify the age and precise location where these calcifications appear, as well as with a quantification of the precise loss of neuronal cells. While we agree that this is important, this is a large study on its own, and we feel this work is

outside the scope of the current manuscript. We have updated Figure 7 to include images of NeuN immunostaining to qualitatively describe the loss of NeuN+ cells in the regions of calcification.

it is particular notable that there is no information about neuronal changes (or lack of changes) in the striatum. Given that HD is generally considered to be a striatal disorder, and that they saw pronounced knockdown in the stratum, it would be useful to include these data- even if they show no change. This would compare directly with the study of Dietrich.

With regards to histology, we chose to focus on the striking thalamic phenotype and loss of tissue in this region. In response to this comment, we reviewed the GFAP and Iba1 histology of our existing tissues. We find no upregulation of GFAP+ or Iba1+ area in the cerebellum, as noted in Dietrich et al and have added this discrepant finding to the results (and Figure S7). While we would like to also examine the striatum, our sections were focused on thalamus and don't adequately capture enough striatal volume to accurately measure GFAP or Iba1 immunoreactivity there. We have noted that we did not quantitate this region in the text.

There is little discussion about why Huntington knockdown should affect the thalamus. It is implied that the changes are specific to the thalamus, but they do not show data from other regions, e.g. stratum or cortex, which would make the specificity of the thalamic changes more convincing.

We thank the reader for making this point. We agree that the fact that calcification is limited to the thalamus is strange and note in the discussion that basal ganglia calcification is a known phenomenon in other human diseases, so perhaps the phenotype seen here is related. With regards to thalamic specificity, we have shown a sagittal image in Fig 5A which demonstrates the striking phenotype is limited to the thalamic area and have updated the figure to have an inset showing this at higher magnification.

Leakiness of the vectors is a concern, and although the authors have suggested that each model represents the effective knockout of either CNS or peripheral HTT, their interpretation of behavior in the mice in this context is not very convincing.

We thank the reviewer for noting this confusing element. We chose to not measure motor behavior in the UBC-cre line, which is quite leaky in the CNS, as we felt that any readout would be severely confounded. The UBC-cre mice (both vehicle- and tamoxifen-treated) had reduced Htt expression in the CNS throughout development, so measuring and interpreting behavior in these mice would not offer clarity on the effects of adult Htt KO that we sought to interrogate.

Since the thalamus was the only region that showed any pathology, leak assessment for this region should be included. It is not clear why this was not included.

We agree that we should have collected this tissue for leak analysis, however we failed to collect this region separately from the microdissection and unfortunately can't recreate this cohort for the submitted manuscript. Our future studies will precisely quantify this, as well as utilize other methods of HTT knock-out, to avoid the confounds of the cre-loxp system.

Minor comments.

Many of the Figures are not well labeled and of low quality. The authors should review them to make them clearer. For example,

There is no key for symbols in Fig 1A.

X axis titles are needed for many panels in most figures, e.g. Figs 1B, 1C, 2A, 3A, 4A, 5B, 6C, D etc.

Statistics indicated in many panels, but details are not included in the fig legends for many figures.

We thank the reviewer for these in depth notes about the figures, we have addressed each of these including the addition of axis labels and symbol legends, where appropriate. With regards to statistics, we have added in symbols to indicate significance, where appropriate, as well as the location of a full list of statistical comparisons available in the supplemental tables.

Figure 5 is supposed to show lack of pancreatic pathology, but most of the illustrated sections show acinar cells, not islets. Better representative images should be chosen to include both and then labeled clearly. Data from pancreas mass should have been included. They said that the pancreas is difficult to dissect, however many other investigators manage this. In the absence of mass data or clear histology, to show there was no pancreatitis, they could have measured Serum amylase, serum lipase, pancreas trypsin activity (as done in Wang et al.)

We thank the reviewer for these suggestions. A qualified veterinarian completed the dissections and noted that the mass and gross appearance did not appear altered - whereas Wang et al noted gross abnormalities. We then had a separate veterinary pathologist score H&E slides, with a particular focus on acinar cells, which were the cells of focus that showed major degeneration and loss of zymogen granules in Wang et al. There was no degeneration or loss of zymogen granules observed. Due to this, we decided not to follow up with additional plasma assays and no longer have the capability with the plasma left on hand. To improve image quality, we have updated the images and added scale bars.

Figure 6 needs labeling.

Fig 7 A needs labeling. Which part of the thalamus is shown? Was there an increase in GFAP throughout the region? It would be useful to see this at a resolution similar to that shown in Fig 6.

We have added additional labeling to Figure 6, as well as adding a low-magnification image in Fig 7. to give readers a better sense of where these images came from. Additionally, we have added scale bars and magnified insets to both Fig 6 and 7 for better cellular resolution.

Details of mice missing from the methods, e.g. when tamoxifen treatment was started. They describe treatment as being lowering at 2 months -this should really be from 2 months since their first time point was 3 months.

We thank the reviewer for noticing this omission and have included the age of tamoxifen initiation (2-months) in both the results and methods section.

Is the balance beam test validated? If so, provide a reference.

We thank the reviewer and have included a reference to this assay in the Methods section describing the protocol in detail:

“Luong TN, Carlisle HJ, Southwell A, Patterson PH. Assessment of Motor Balance and Coordination in Mice using the Balance Beam. J Vis Exp. 2011;(49).”

Is not clear why, given this was a replication study, that they did not use the same mice as either of the previous studies. They should explain the rationale for using different breeding strategies.

We thank the reviewers for bringing up this point - as noted in an earlier comment, we have clarified in the text that we used the same model as the Wang et al study, which is a modified version of the Deitrich et al study. We feel this is an improvement that brings both studies to a directly comparable set of experiments.

line 56 there is no need to abbreviate loss of function.

line 57 gnomAD need spelling out in full.

line 70 a reference is needed for lack of effect of 50% HTT expression.

We have addressed these notes and added a reference to Ambrose et al (1994) that describes a translocation in which a phenotypically normal patient had one copy of HTT.

Reviewer #2 (Comments to the Authors (Required)):

We thank the reviewer for their careful reading of our manuscript, which they accurately summarize below. Our responses to each of the concerns they raised are below in bold, interspersed with their original comments.

The development of huntingtin-lowering therapies has been a major focus in recent years and a wide range of huntingtin lowering treatments are in clinical trials. However, the levels to which the huntingtin protein can be lowered without adverse consequences is unknown and a potential confound for many of these approaches. It has been known for close to 30 years that the complete loss of huntingtin causes severe developmental symptoms in mice and more recent data has shown that this is also true in humans. However, it is the extent to which huntingtin can be lowered in adulthood that is relevant to the design of treatments that will be safe in the clinic. This question has previously been addressed by conditional knockout studies in mice. One study found that the early induction of global HTT loss caused fatal pancreatitis, but that lowering HTT at later ages was benign. Another study found that postnatal HTT loss was associated with subcortical calcification and neurodegeneration but did not report fatal pancreatitis.

In this study, the authors have set out to better understand the risks posed by the widespread inducible loss of HTT. They utilised two inducible tamoxifen Cre lines, each under the control of ubiquitous promoters. This was important as these lines exhibited complementary levels of leakiness; broadly speaking, the UBC promoter was leaky in the brain but not the periphery, whereas the CAG promoter was leaky in the periphery but not the brain. They characterised the pattern of wild-type HTT loss throughout the brain and periphery when conditional knock-out was initiated at 2 months of age. They found that widespread loss of HTT at 2 months of age led to a wide range of phenotypes, including subcortical calcification, but did not result in acute pancreatitis or histological changes in the pancreas. Importantly, they found that HTT loss was accompanied by a robust and sustained increase in the levels of neurofilament light chain (NEFL), a peripherally accessible biomarker that is widely in HD clinical trials. They confirmed that the complete loss of HTT in mice is associated with pronounced risks, including progressive subcortical calcification and neurodegeneration.

In all cases the data strongly support the conclusions of the paper.

This well-conducted and well-powered study is extremely important and will be of great interest to all researchers and clinicians in the Huntington's disease community.

There are some minor comments.

Gene and protein nomenclature for huntingtin should be HTT (*italics*) for human gene, Htt (*italics*) for mouse gene and HTT for the human and mouse protein.

The gene and protein symbols for neurofilament light chain as approved by the HUGO Nomenclature Committee are NEFL (*italics*) and NEFL respectively (see Caron et al. 2022 Brain Comm. Fcac309), although there are many aliases for the protein.

We have updated the manuscript to ensure that HTT/HTT/Htt are written in proper format and have converted NfL to NEFL.

Introduction

Page 4, line 89: typo 'at beginning at'

Thank you—fixed

Results

FigS6. It would be helpful if symbols were defined in the figure legend.

Thank you—fixed

Fig.3C legend. Could the authors add quantification and p values to the statement 'Httfl/fl;CAG-Cre mice travelled slightly less distance than control mice'

Thank you—fixed

Fig 5A the comparative images of pancreatic histology do not appear to have been presented at the same magnification. Could the authors show higher magnification images and point out the aspects of histology mentioned in the text e.g. zymogen granules.

We've updated this panel to ensure images are at equal magnification and have added scale bars. We now include a note in the figure legend describing that these are primarily acinar cells and that they all appear full of zymogen granules.

Fig 6. Could the authors define Tfrc in the legend?

Updated to indicate transferrin receptor (TFRC)

Behavioral analysis - page 12 line 349. Table 4 should be Table S1

Fixed

Materials and Methods

The authors do not mention the mouse strain background that they are using in their studies and whether this is the same background as was used by Dietrich et al. and by Wang et al.

We thank the reviewer for this comment. We have added a note that we used mice in the C57BL/6 background (B6), confirmed with a 384 SNP panel. We also noted that the background strain is listed as B6 in Deitrich and is likely B6 in Wang et al, however neither of these groups indicated whether they confirmed the Htt flox line received from Dr. Zeitlin was backcrossed to B6 congenicity.

Reviewer #3 (Comments to the Authors (Required)):

We thank the reviewer for their careful reading of our manuscript, which they accurately summarize below. Our responses to each of the concerns they raised are below in bold, interspersed with their original comments.

Huntingtin, the protein affected in Huntington's disease has many well-described functions and the consequences of lowering this protein using various methods including tamoxifen Cre inducible systems has been described. In Huntington's disease (HD), typically one allele has the mutation. Non-allele specific lowering of huntingtin is an approach being evaluated for treatment of HD in mice and in patients. This replication study addresses the effects of lowering huntingtin in C57Bl6 mice using two inducible systems reported by others. In the present work huntingtin was reduced at 2 months and the consequences were examined at 14 months. This well-written and nicely illustrated study presents a comprehensive behavioral, molecular and histological picture that replicates some of the adverse phenotypes reported in the two prior studies. The evidence that lowering huntingtin can signal an elevation in plasma and CSF levels of Neurofilament protein, a marker of compromised axonal integrity, is a significant new contribution. Overall, this work adds considerable value because of the importance of this issue for future clinical therapy, the new data reported, and the care the authors took in detailing the caveats associated with the experimental models.

Scale bars need to be added to some of the figures.

Thank you. Scale bars have been added to all figures with images.

Fig 7 An inset to show cellular detail for GFAP at least would be helpful. Are the dark circular areas cross sectioned blood vessels?

Inset added - the dark circular areas are the large calcified deposits as shown in figure 5.

Adding P values or asterisks to box plots for significant comparisons would be very helpful.

We have added asterisks to boxplots as well as a complete set of statistical data to the supplemental files

Hildich-McGuire et al., 2000 is another good paper to site.

Added into the iron section

June 14, 2024

RE: Life Science Alliance Manuscript #LSA-2024-02571R

Dr. Jeffrey B Carroll
University of Washington
Department of Neurology
HMC #359660
325 9th Ave
Seattle, WA 98104-2499

Dear Dr. Carroll,

Thank you for submitting your revised manuscript entitled "Global Huntingtin Knockout in Adult Mice Leads to Fatal Neurodegeneration that Sparing the Pancreas". We would be happy to publish your paper in Life Science Alliance pending final revisions necessary to meet our formatting guidelines.

- please be sure that the authorship listing and order is correct
- please add an Abstract and a Summary Blurb/Alternate Abstract to our system
- please add the Twitter handle of your host institute/organization as well as your own or/and one of the authors in our system
- titles in the system and manuscript file should match
- please add an Author Contributions section to your main manuscript text
- please move your main, supplementary figure, and table legends in the main manuscript text after the references section
- we encourage you to revise the figure legends for Figure S1 such that the figure panels are introduced in alphabetical order
- please add callouts for Figures 3B; 4A-B; 5B and S5A-C to your main manuscript text

FIGURE CHECKS

- please add scale bar and size to Figure S6

LSA now encourages authors to provide a 30-60 second video where the study is briefly explained. We will use these videos on social media to promote the published paper and the presenting author (for examples, see <https://docs.google.com/document/d/1-UWCfbE4pGcDdcgzcmiuJl2XMBJnxKYeqRvLLrLSo8s/edit?usp=sharing>). Corresponding or first-authors are welcome to submit the video. Please submit only one video per manuscript. The video can be emailed to contact@life-science-alliance.org

A. FINAL FILES:

B. MANUSCRIPT ORGANIZATION AND FORMATTING:

Sincerely,

June 24, 2024

RE: Life Science Alliance Manuscript #LSA-2024-02571RR

Dr. Jeffrey B Carroll
University of Washington
Department of Neurology
HMC #359660
325 9th Ave
Seattle, WA 98104-2499

Dear Dr. Carroll,

Thank you for submitting your Research Article entitled "Global Huntingtin Knockout in Adult Mice Leads to Fatal Neurodegeneration that Spares the Pancreas". It is a pleasure to let you know that your manuscript is now accepted for publication in Life Science Alliance. Congratulations on this interesting work.

DISTRIBUTION OF MATERIALS:

Again, congratulations on a very nice paper. I hope you found the review process to be constructive and are pleased with how the manuscript was handled editorially. We look forward to future exciting submissions from your lab.

Sincerely,
